# Dried Blood Spot in Toxicology: Current Knowledge

## Agnieszka Niemiec

Department of Forensic Medicine and Forensic Toxicology, Medical University of Silesia, Poniatowskiego 15, 40-055 Katowice, Poland; rektor@sum.edu.pl or niemiec.agnieszka@interia.pl; Tel.: +48-53-167-27-50

**Abstract:** Dried Blood Spot (DBS) is becoming very popular in various medical fields, especially in toxicology. Nowadays it is commonly used in newborn screening for inherited or congenital diseases. This paperwork is based on a review of available literature. DBS is simple and rapid, it does not require trained medical staff to collect the samples. Specimens can be easily and safely transported to the laboratory. DBS provides an opportunity for roadside testing and rather quick results. Venous blood spot, collected from a finger or a heel, is put on the special paper card, which can result in a different distribution of blood and concentration of detecting substances. Marking multiple substances from one spot is extremely challenging, but due to further advancements in this area, it is only a matter of time until it becomes possible and all the disadvantages vanish. DBS is certain to develop and become even more worldwide used.

**Keywords:** dried blood spot; dried blood spots; dried blood filter; dry blood spot; toxicology; forensic; filter paper; DBS





## 1. Introduction

Dried Blood Spot (DBS) is a technique that involves collecting small samples of venous blood (plasma or serum are applicable as well), usually from a finger, toe, or heel, onto an absorbent filter paper. Whatman® 903 paper is frequently used since it is made from pure cotton fibers [1,2]. Exceptionally, at the crime scene, other materials than paper are used. After the drop of blood is air-dried, it can be transported to a laboratory and analyzed. The paper card must be stored in a gas-impermeable zipper bag along with desiccant packets. The bag itself is put into the freezer. In the laboratory, all the spots are punched out and eluted. After the formation of hemolytic supernatants and centrifugation, the spots are ready to be analyzed using immunoassays or molecular techniques (GC-MS, LC-MS/MS) [3]. DBS is widely used for the detection of HIV, hepatitis B, or C; however, it is also vital in newborn screening for inherited or congenital metabolic disorders [4]. Recently, this method is gaining popularity in the field of toxicology, where it can contribute to the assessment of substances of abuse, toxins, and trace elements. This paper focuses on the aspect of using DBS in toxicology.

## 2. Materials and Methods

This paper was prepared through a review of available literature found using keywords such as DBS, toxicology, filter paper, screening, and blood collecting. There were several websites used for this search: PubMed, Science Direct, Wiley Online Library, and Frontiers. All articles used were published in prominent scientific journals. Most of them are listed below in the References section, whereas some did not reveal anything new or important enough to be acknowledged as a reference. I have read over thirty articles in total to acquire knowledge of this topic.

## 3. Results

### 3.1. History

DBS was discovered in 1913 by Ivar Bang, who used it for glucose concentration monitoring in rabbits. Half a century later, in 1963, Guthrie and Susi managed to prove blood sampling useful in screening for phenylketonuria. In the 1970s, DBS was implemented in serological surveillance and used to diagnose syphilis and detect antibodies for mumps and measles. Due to the increase in HIV morbidity, DBS has attracted attention at the beginning of the year 2000 and successfully helped to monitor HIV infection. Nowadays it is used commonly in various fields such as medicine, pharmacy, and new technologies [5,6].

### 3.2. Drugs Analysis

DBS may be significant for detecting drugs and help pharmaceutical concerns in conducting research. Multiple substances can be measured in DBS: benzodiazepines, Z-drugs (zolpidem, zopiclone), opiates (6-monoacetylmorphine, morphine, codeine, hydromorphone, hydrocodone, oxycodone, noroxycodone), tramadol, methadone, buprenorphine, fentanyl, ketamine, and their respective metabolites [7] (and references therein). Moreover, there are reports that it is possible to mark the level of ethyl glucuronide that serves as an alcohol abuse marker [7] (and references therein). Analyzing drugs with DBS serves two main purposes. Firstly, it can detect drug abuse in adults or measure the level of medication (TDM—Therapeutic Drug Monitoring) and improve a follow-up. Secondly, it can evaluate the exposure to the drugs before birth in newborns. Both therapeutic and illegal ones. The aforementioned TDM used with DBS has proven to be highly effective in monitoring busulfan in children, who had to undergo a hematopoietic stem cell transplantation [8]. In addition, a study by Hahn et al. showed that topiramate TDM is also possible and highly beneficial [9]. Finally, a study by Duthaler et al. contains good results of therapeutic drug monitoring of antiretroviral drugs in resource-poor regions using the DBS technique [10].

#### 3.2.1. Substances of Abuse Marker

Detecting substances of abuse via DBS is still under a lot of research. However, there have been numerous reports that it is possible to detect cannabinoids, cocaine, and its metabolites as well as opiates, such as morphine, non-opiate opioids, amphetamine, and its analogs and gamma-hydroxybutyric acid [7]. Morphine may cause some issues since it is a heroin (illicit drug) or codeine (licit drug) metabolite. Marking 6-Monoacetylmorphine is a solution because it is one of the active heroin derivatives [11]. The vast majority of these drugs have a short half-life, but DBS has a stabilization effect on them so they can endure for a longer period [7]. It is also worth mentioning that a recent study by Moretti et al. describes a successful analysis of a few psychoactive substances and their metabolites in postmortem blood [12]. However, research conducted by Patteet et al. also confirmed that DBS is vital in therapeutic drug monitoring of antipsychotics, but it excluded olanzapine and N-desmethylolanzapine [13].

Driving under influence of drugs (DUID) has been discussed in the DRUID project. The role of DBS in this matter has been evaluated, but only to some extent [7]. Oral fluid, which is widely used for DUID testing is not a perfect method. There are some controversies, whether the results are comparable to the ones obtained via usual blood testing. What is more, they can be falsified by different conditions such as the use of mouth wash [14]. DBS seems to be more reliable and its sampling is quick and does not require any extra knowledge or experience so it can be collected by an officer during a routine control.

Additionally, there was a study by Simões et al. performed to check the use of DBS along with UPLC-MS/MS technology in the field of forensic toxicology. As a result, DBS was proven to be useful in the detection of illicit drugs [15].

#### 3.2.2. Trace Elements Detection

There is a wide range of elements confirmed to be detected in DBS: Pb, As, Ba, Be, Bi, Ca, Cd, Co, Cr, Cs, Cu, Fe, Hg, K, Li, Mg, Mn, Mo, Na, Ni, P, Rb, S, Sb, Se,

Tl, V, and Zn [7] (and references therein). There is a possibility of creating a "metallic profile" of an individual to assess the exposure to contamination and take appropriate measures to clean the environment [16]. It may be challenging due to the differences in the distribution of different elements on paper and the possible contamination during collection or transportation.

### 3.3. Newborn Screening

Although DBS in newborn screening is widely used for detecting congenital or inherited diseases, it can also help in assessing the prevalence of tobacco and cocaine in pregnant women [7]. It is performed by detecting benzoylecgonine (cocaine metabolite) and cotinine (nicotine metabolite). Unfortunately, it will not assess the whole pregnancy period, but only the time near the delivery due to the elimination of these substances from the organism. However, the immature liver of newborns functions slower than in adults, so it extends the time for detection of these substances. The first such report was published by Henderson et al., who used redesigned urinary benzoylecgonine radioimmunoassay screening test to assess using cocaine by the mother during pregnancy [1].

Not only measuring elements in adults corresponds with the contamination of the environment. The benzene oxide, perchlorate, organochlorine dichlorodiphenyldichloroethylene (DDE), the PFCs perfluorooctane sulfonate (PFOS), and perfluorooctanoate (PFOA) have been evaluated in newborns with the use of DBS [7].

### 3.4. DBS Testing in SARS-CoV-2 Serology

Nowadays the possibility of detecting SARS-CoV-2 antibodies is being thoroughly studied. Numerous studies have proven DBS effective in antibodies detection with accuracy comparable to serum or plasma samples [17,18]. This method is less expensive and easier, so it can be widely used in the pandemic. It can be used not only to confirm an infection, but also to test vaccine-stimulated antibody response [19].

### 3.5. Advantages of DBS

DBS has several advantages. Firstly, it can be easily collected. It is simple and there is no need for trained medical staff to perform sampling. The extraction procedure is simplified, economical, and cost-effective. There is a possibility for automatization, which would increase the speed of this already rapid process. Secondly, there is a low biohazard risk during transportation. There is no leakage because paper cards with blood are dry so it is safe for the personnel. Safety is also maintained due to the loss of infectivity of some viruses during the drying process. That applies to the HIV-1 and -2 viruses, the human T-cell leukemia/lymphoma virus-I and -II, and the hepatitis C virus [20]. Thirdly, DBS has a stabilizing effect on drugs and it inhibits degradation of the substance [21]. The other advantage of DBS that is worth mentioning is the ability to assess the acute state of the patient. This technique is quick and can provide crucial information about a patient's health, to treat him as fast as possible [20]. Another important advantage is gender neutrality and lack of adulteration issues [22]. The other asset to be enumerated is the ability to keep DBS paper cards as evidence, even when the case is closed and other evidence is discarded. Their storage is uncomplicated and they do not require a lot of space. The downside of this idea is that paper cards cannot be stored forever due to their limited preservation time [7] (and references therein). Therapeutic drug monitoring is a perfect place for DBS. Due to easy sampling and speed, DBS can be performed before the doctor's arrival and show the concentration of certain substances (such as antibiotics, antidepressants, immunosuppressants, and antiretrovirals) in the patient's blood [23]. Liquid extraction or solid-phase extraction requires more blood (around 100–2000 µL) than DBS (10–100 µL), which makes DBS more convenient [21]. Finally, there is an opportunity to perform roadside testing for impaired driving. Collecting blood on the crime scene is reliable and practical [21].

### 3.6. Disadvantages of DBS

DBS may seem flawless, but it has some downsides as well. The drying time is quite long and lasts around 2 h, depending on the conditions such as the type of card and blood volume. Also, the blood can coagulate or lyze, which makes the distribution on the paper card differ [24]. The viscosity of the blood has a similar impact on blood distribution [2]. Then, the results may be disturbed and not reliable [21]. Moreover, detecting some substances may be difficult. Few substances have been measured with the DBS technique. Further research is needed. Due to the small volume of blood collected, it may be impossible to detect multiple drug groups at once. It would require a few DBS samples, so the whole process would be elongated and more complicated [1].

### 3.7. Future Perspectives

DBS has enormous potential for massive development. It is constantly being automatized and new robots are being designed to make DBS less dependent on human beings. Direct analysis technologies are evolving and filter paper is being brought to perfection. Probably, it will be good enough to provide the same blood distribution within the spot and preserve the substances for a longer period. Standardization of the whole process may contribute to the increase of results reliability. Hopefully, in the future, there will be extensive research of postmortem samples, that are hemolyzed and putrefied [25]. Furthermore, there are derivatives of the DBS technique such as perforated DBS (PDBS), bilayer DPS card, and Hemaspot technology [26]. They may replace usual DBS and create new possibilities for substances detection. Especially, the pharmaceutical sector is bound to use DBS even more and it may improve their clinical studies [27]. Also, worldwide preparing DBS kits divided into panels for metabolic, hormonal, and cardiovascular disorders can speed along the diagnostics.

## 4. Discussion

DBS is a developing technology that provides wide possibilities for various medical fields. Its simplicity makes it available for nearly everyone. It is safe and cost-effective. Stabilization of the substances in blood on paper cards elongates the time before detection and prevents de novo formation of substances, which can change the results. This factor is also vital in keeping the evidence after the case is closed since paper cards do not require much place. The storage and transportation of the biological material are safe and rather fast. Results come quickly so they can help treat the patient in the right way. Therapeutic drug monitoring is easier and more accessible due to the DBS. Roadside testing (especially for alcohol) with DBS can dodge results' disturbances such as a mouth wash. It is already commonly used in newborn screening and it is being introduced to pharmaceutical companies, which benefit a lot from this technology. Creating a "metallic profile" of an individual makes DBS contribute to the environment as well. The aforementioned disturbances do not seem to be impossible to overcome. Further research is needed to implement DBS on a wider scale, but it looks very promising for all fields, especially toxicology.

**Funding:** This research received no external funding.

**Institutional Review Board Statement:** Not applicable.

**Informed Consent Statement:** Not applicable.

**Acknowledgments:** Koło Naukowe Medycyny Sądowej SUM.

**Conflicts of Interest:** The authors declare no conflict of interest.

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
