# Peer review of "Dried Blood Spot in Toxicology: Current Knowledge"

_separations, doi:10.3390/separations8090145_

Round 1

Reviewer 1 Report

It is necessary to deepen the research on the subject (DBS) to ensure that the text becomes a scientific article.

Paragraph 2 needs to be more detailed, specifying what kind of literature review has been done, for example, a systematic review listing the keywords used in search engines and the number of scientific articles viewed and processed.

On page 2 line 65, insert the following reference:
Borriello R, Carfora A, Cassandro P, Petrella R, 2015. Clinical and Forensic Diagnosis of Very Recent Heroin Intake by 6-acetyl morphine Immunoassay Test and LC-MS / MS Analysis in Urine and Blood. Annals of Clinical & Laboratory: Vol. 45, No 4: 414-418. ISSN: 0091-7370.

On page 2 line 85, insert the dot.

Please, check the formatting of the authors for references 1, 5, 7, 8,13 and 15

Author Response

1) I've deepened my research a little bit, but I do not know in which direction I should go. 

2) Thank you, I had no idea what to put in the 2nd paragraph, so I have added some information just like you have suggested.

3) I have added the reference.

4) I have inserted the dot.

5) I have corrected the author references (now there are full surnames).

Reviewer 2 Report

This paper reviews Dried Blood Spot Analysis, an important and continually emerging area for medical/forensic applications. My biggest concern is that the authors heavily cite reference [5] which is another review article on ostensibly the exact same topic (Dried Blood Spots in Toxicology: From the Cradle to the Grave? Critical reviews 177 in toxicology 2012, 42, 230–243). It is cited two times in Section 3.2., three different times in Section 3.2.1, at least twice in 3.2.2 and 3.3, etc...

Because reference [5] was published several years ago (i.e., 2012), it would be much more useful if the authors found citations since this 2012 paper to accurately capture more recent developments in the field. Additionally, I strongly encourage the authors to use primary studies for their citations rather than review articles whenever possible. Further if using a review article for a citation, the text should read '[5], and references therein'. However this should be done sparingly.

Although the article reads fine, there are a few grammatical issues that needs to be fixed in a revision. For example, there are mixups of singular versus plural in the Abstract, places where 'DBS' should be used instead of 'Dried Blood Spot' (e.g., Line 42), and incorrect word choice-e.g., 'sachet' in Introduction is incorrect.

Author Response

I have added more references to my paper (mostly newer ones, from 2020 and 2021) and added the text 'and references therein' in some of my citations. I have also tried to correct grammatical issues and taken care of mixups of singular vs. plural. Finally, I have added a small paragraph about DBS in COVID-19 testing, because it is a brand new application of DBS, but, in my opinion, it does not strictly belong to the toxicological field.

Round 2

Reviewer 1 Report

I think that in this version, the manuscript is suitable for the publication